# Design and Evaluation of an Outdoor Exercise Program for Pediatric Cancer Survivors

**DOI:** 10.3390/children9081117

**Published:** 2022-07-27

**Authors:** Christopher Blosch, Arno Krombholz, Ronja Beller, Gabriele Gauß, Dirk Reinhardt, Miriam Götte

**Affiliations:** 1Department of Pediatric Hematology/Oncology, West German Cancer Centre, Clinic for Pediatrics III, University Hospital Essen, 45147 Essen, Germany; christopher.blosch@rub.de (C.B.); ronja.beller@uk-essen.de (R.B.); gabriele.gauss@uk-essen.de (G.G.); dirk.reinhardt@uk-essen.de (D.R.); 2Faculty of Sport Science, Ruhr University, 44801 Bochum, Germany; arno.krombholz@rub.de

**Keywords:** childhood cancer, physical activity, cancer survivors, green exercise, promoting, inhibiting, outdoor activities

## Abstract

Exercise programs for young people after cancer are not part of regular oncological care. This study describes and evaluates a regional outdoor exercise program and presents data with regard to the promoting and inhibiting factors for participation among pediatric cancer survivors. Exercise options, number of participants, and the cohort were evaluated descriptively for one year. A self-developed questionnaire was used to evaluate satisfaction, mood, motivations, and barriers to exercise. Overall N = 26 survivors (14.6 ± 5.5 years) participated in at least one activity in 2019 including try-out days (N = 10) and active weekend camps (N = 2). No adverse events occurred in 302 physical activity hours. Twenty-one survivors participated in the survey. The largest motivational aspect to participate was “to try out a new sport” (83.9%). Survivors reported “good mood”, and ‘being happy’ after exercising. The largest barrier was concern about ‘not being able to keep up with others’ (38.1%). Around one-third (try-out day) and 50% (active weekend camp) of survivors did not feel confident to continue exercising outside the supervised exercise oncology program. This survey shows high enthusiasm for this exercise program with different outdoor activities and suggests that similar interventions may be accepted by this population.

## 1. Introduction

Sports and physical activity are important requirements for the physical, social, and mental health of children, adolescents, and young adults [1]. However, children diagnosed with cancer are physically less active than their healthy peers during and even after treatment [2,3]. This seems to be attributed to disease- and treatment-related symptoms or side effects (e.g., fatigue, nausea, pain, immunosuppression, or neuropathy) [4,5,6]. Consequences of inactivity are skeletal muscle atrophies and limited physical performance, which lead to a further reduction in quality of life, physical activity, and autonomy [7].

There is growing evidence for positive effects of exercise interventions for childhood cancer survivors, such as alleviated treatment symptoms, reduced fatigue, and improved physical performance [8,9]. However, the international prevalence of exercise intervention programs and concepts for children diagnosed with cancer is low [10]. The Network ActiveOncoKids (NAOK) aims to implement pediatric exercise oncology as usual care for children, adolescents, and young adults with cancer in Germany [11] and an increasing number of hospitals are offering exercise programs based on NAOK support, counseling, guidelines [12], and policy changes such as collaborations with health insurance companies. Most of the existing interventions take place during therapy and the number of aftercare interventions is very low. The majority of existing interventions involve traditional types of exercise such as endurance and resistance training or supervised exercise programs organized in exercise groups [13,14]. However, such interventions may not appeal to a younger cohort who, in qualitative research, expressed a desire for more social exercise programs involving outdoor activities like climbing, surfing, or kayaking [15].

Previous research has shown that outdoor exercise interventions increase physical activity in children, adolescents, and young adults with different chronic diseases and disabilities such as cancer or autism [15,16,17]. Participants particularly benefit from the special characteristics and settings of the respective sport. Outdoor exercise has been shown to be highly beneficial to general health, especially in the area of emotional well-being in different populations and age groups [18]. Various studies have shown that outdoor exercise has the potential to strengthen self-confidence [19] and reduce anxiety and stress [20] in adults. Especially children in a school context whose participation in sports facilities is associated with fears and insecurities seem to benefit from outdoor exercise [21]. In addition, outdoor exercise may have a positive influence on patients′ body perceptions in cancer survivors. Studies show direct correlations between reduced body perception and low self-esteem, especially in preadolescents and adolescents [22]. Outdoor exercises are individual and noncompetitive sports. Participants have the opportunity to manage legitimate outdoor challenges in accordance with their individual resources and gain individual experience without having to compete with healthy peers. Participation by pediatric cancer survivors in outdoor exercise interventions might have the potential to provide positive experiences for survivors by reviving their self-efficacy lost to cancer, regaining confidence, and encouraging them to lead a healthier and more active lifestyle [15].

To close the gap between acute cancer treatment and an active survivorship, the NAOK Ruhr Center, a regional center of the nationwide NAOK, implemented outdoor exercise interventions as an active survivorship program in 2016.

The main objective of this pilot study was to describe and evaluate the concept of this outdoor exercise program for childhood, adolescent, and young adult cancer survivors. This study may provide information for those interested in designing and implementing programs for adapted physical activity during aftercare, as well as for others looking at the feasibility of outdoor exercise for childhood cancer survivors.

Practical experience showed general feasibility of an outdoor exercise program among survivors, but partly low participation rates for a larger potential cohort. Therefore, secondary outcomes of this study were to evaluate satisfaction of the program exemplary for the year 2019, as well as identifying the mood, motivations, barriers, wishes, and information about further exercising. Findings will help to better adapt the offers to the needs of childhood cancer survivors and to increase the participation rates and the exercise benefits of pediatric patients and survivors.

## 2. Materials and Methods

### 2.1. Programme and Organization

The regional center of the Network ActiveOncoKids (NAOK) is jointly coordinated by a large pediatric oncology university hospital and a nearby faculty for sport science. It is a local association of the national Network ActiveOncoKids that aims to provide access to exercise programs for children, adolescents, and young adults during and after cancer treatment all over Germany (http://www.activeoncokids.de accessed on 26 May 2022) [11]. The main objective of the regional center is to establish and expand exercise programs for children and adolescents after cancer treatment in the Metropole Ruhr area.

The exercise program is primarily focused on outdoor sports like windsurfing, sailing, kayaking, rowing, stand-up paddling, scuba diving, and climbing. The different sports are offered in form of several try-out days and active weekend camps throughout the year. At the try-out days, participants have the opportunity to experience and learn new kinds of sports. They can gain further experience in the respective sports by participating in the try-out days several times or by joining an active weekend camp. Depending on the sport, the weekend camps include two to three practice sessions per day, as well as theoretical lessons. At the end of the active weekend camps, it is even possible to acquire licenses in some sports (scuba diving, sailing, and windsurfing).

The exercise options are organized and implemented by a multi-professional team from the acute cancer hospital and the sports institute. In terms of content, the programs consist of a collaboration between the cooperation partners. The acute hospital contributes expertise from the fields of exercise oncology interventions and pediatric oncology, maintains a close exchange with the treating physicians or tumor orthopedists, and ensures contact with the patients. The sports institute and qualified movement instructors provide expertise in the areas of the didactics and methodology of the sports offered, resources of sports equipment, networks for external sports options, and the involvement of students (Figure 1). Prior to a patient′s participation, an interdisciplinary approach is used to discuss the assessment of potential risks and anticipated benefits with the treating physicians and therapists. Each session is planned according to the needs of the survivors (e.g., diagnoses, handicaps, and late effects). The regional center of NAOK responds individually to the different disabilities of the participants, e.g., by adapting the framework conditions or materials and exploring ways of exercising with a disability. Sports scientists and instructors provide one-on-one instruction whenever needed to accommodate survivors with physical limitations and to make it accessible to everyone.

### 2.2. Sports Programme in 2019

In 2019, the outdoor exercise program included try-out days for windsurfing, sailing, kayaking, rowing, stand-up paddling, scuba diving, and climbing, as well as active weekend camps for scuba diving and climbing and one for water sports, in which several sports could be experienced (windsurfing, sailing, kayaking, and stand-up paddling).

### 2.3. Participants’ Inclusion and Recruitment

All patients with a history of cancer, older than the minimum age for the respective sport (minimum 5 years) who were treated in one of the two collaborating acute hospitals were offered to attend the exercise program. This program is open for young adults, too, although it was mainly designed for children, adolescents, and young adults. Written registration and medical clearance were required prior to participation. Patients were recruited by clinical exercise physiologists and pediatricians during treatment or at follow-up appointments in the outpatient hospital. Further advertising includes posters and brochures on the ward or a homepage. Patients were encouraged to register together with a sibling or a friend. This “buddy concept” was intended to lower the individual barrier for participation and to facilitate the continuation of exercising in their social environment.

### 2.4. Survey Design and Procedure

A self-developed questionnaire based on previous qualitative research [23] was used to anonymously determine satisfaction with the exercise interventions and to evaluate motivations and barriers to participation in sports. The software EvaSys was used to prepare the questionnaire and conduct the online survey. All participants who took part in at least one activity of the aftercare program in 2019 were eligible to participate. All potential subjects were approached in an email to take part in the retrospective online survey via a link. Reminder emails were sent after one and two weeks. Parents of participants aged 14 years or younger were asked to support their children in answering the questionnaire.

The questionnaire consisted of five content items: (a) general information (four questions), (b) try-out days, (c) active weekend camps, (d) motivations/barriers (11/14 questions), and (e) wishes/comments. Items of try-out days and active weekend camps included questions about personal well-being (13 questions) and further practice of the sport (five questions). The majority of questions were statements that participants could rate using a five-point Likert scale (from ‘strongly disagree’ to ‘strongly agree’). Other types of answers included yes/no, free text answers, or other statements on a five-point scale. In the motivation scale, all questions began with the same statement (‘I participated in the NAOK sports program because...’). The continuation of the statement showed different expressions (e.g., ‘I wanted to try a new sport’. This was followed by another five-point Likert scale (from ‘disagree’ to ‘agree’). The barriers scale included several statements, such as ‘I have no motivation in general’ or ‘My physical performance is too poor’, and participants were asked to rate them using the same five-point Likert scale (from ‘disagree’ to ‘agree’). At the end of the survey, participants had the opportunity to express wishes or comments in free text form. The questionnaire is available in Appendix A.

Ethical approval was granted by the ethic committee of the University of Essen (22-10651-BO). Participation in the survey was voluntary, and subjects were informed in advance that they were giving consent by participating. Before generally participating in one of the sport activities, all subjects provided informed written consent for their data to be used anonymously for statistical analysis regardless of participating in this survey.

## 3. Results

### 3.1. Participants and Activities in 2019

In total, 26 participants (14.6 ± 5.5 years, gender = 14 male, 12 female) took part in the outdoor exercise program in 2019 with different cancer entities and disabilities (Table 1).

Overall, 19 activities were planned for the 2019 program. Of these, 16 were try-out days and three were active weekend camps (Table 2). Seven activities did not take place. Reasons for cancellation were: less than three registered participants (n = 5), thunderstorm (n = 1), and illness of the instructors (n = 1). In total, 97 participations occurred in the various activities, 53 by cancer survivors and 44 by buddies. Of these, 76 participated in the try-out days (survivors n = 41, buddies n = 35) and 21 in the weekend camps (survivors n = 12, buddies n = 9) (Table 2). Try-out days lasted from 2.5 to 5 h. Prior to each practical session, there was a theoretical introduction to the respective sport in which topics such as material science and rules of conduct and safety were covered. Due to additional time for dressing, meetings, or breaks for eating and drinking, the active hours were between 1.5 and 2.5 h per session. For active weekend camps, the active hours were between 6 and 8 h. No adverse events were registered in a total of 302 active hours.

### 3.2. Survey Results

Twenty-one of the 26 individual participants eligible for the questionnaire accepted the invitation to participate (80.8%) (Figure 1). The questionnaire was completed by the participants alone (52.4%) or together with their parents (47.6%). One-third of the participants were each between 8–12, 13–17, and over 18 years old. Most participants became aware of the offer ‘through the staff of NAOK’ (47.6%). Further recruitment pathways can be seen in Figure 2.

The most comment motivations for participating in the NAOK program were ‘I wanted to try out a new sport’ (83.9% agree or somewhat agree), ‘I was allowed to bring a buddy’ (66.6%), and ‘I wanted to improve my physical well-being’ (57.2%) (Figure 3). If the participants are divided into subgroups according to their age group (8–12, 13–17, and over 18 years), differences can be seen with regard to this content item. In the age group 8–12 years, 100% agree with the answer ‘My family recommended it to me’. Number one answer of age group 13–17 years was ‘I wanted to try out a new sport’ (100% agree). The answers of participants over 18 years were more variable, but also with the most frequent mention of trying out a new sport (71% agree).

After the try-out days, participants reported that they were in a good mood (100%), satisfied (94.7%), happy (89.5%), and self-confident (89.5%) (Figure 4). Moreover, they had no pain (100%) and did not feel listless (94.7%) or disappointed (94.7%). After the active weekend camps, participants stated that they were happy (100%), satisfied (92.9%), and in a good mood (92.9%). Furthermore, they did not feel listless (92.9%), disappointed (92.9%), had no pain (85.7%), or were not glad that it was over (85.7%). Overall, 89.5% and 92.5% of try-out day and active weekend camp participants, respectively, rated the exercise programs as good to very good.

In the questions about the further exercising, 84.2% (try-out day) and 85.7% (weekend camp) of participants reported about progress in the respective sport (Figure 5); 63.1% (try-out day) and 50% (weekend camp) want to practice the sport also beyond the supervised offer of NAOK Ruhr. The statement that there is a possibility to practice the sport near home was agreed to by 36.9% (try-out day) and 42.8% (weekend camp) of participants. However, 36.9% (try-out day) and 50% (weekend camp) of participants reported that they do not have the confidence to perform the sport without the support of the NAOK Ruhr Center.

The main barriers of doing exercise outside the NAOK program were ‘Concerns about not being able to keep up with the others’ (38.1% agree or somewhat agree), ‘he burden of school/studies/work’ (33.3%), and ‘My physical performance is too poor’ (23.8%) (Figure 6).

Swimming, trampoline, dancing, and BMX were mentioned as wishes for future sports. Swimming was especially mentioned several times. Furthermore, the desire for a weekly exercise program was expressed.

## 4. Discussion

Several important findings were obtained from the study that provide information for the implementation of outdoor exercises as an aftercare program in pediatric cancer survivors. While different studies have shown that survivors have problems returning to physical education at school or sports clubs after treatment [24,25,26], exercise programs that specifically address the needs of survivors are urgently needed. The reasons for the absence of aftercare programs for this cohort are attributed to various barriers. Often there is a lack of expertise, existing structures, personnel, or financial resources [27]. The fact that the programs are not part of standard care also contributes to the lack of support.

The NAOK Ruhr Center initiative is intended to offer children, adolescents, and young adults their first experience of new sports at a time when a return to the old sports often does not take place because of existing physical limitations or insecurities. In our cohort, 38.5% of the participants had at least one severe late effect. This is a higher proportion than the average for cancer survivors of about one-quarter [28]. It seems, therefore, that our offer has specifically reached those with high needs. This targeted offer might especially enable this target group to participate because, in contrast to regular sports clubs, it provides adapted materials, specially qualified and experienced instructors, and low barriers to participation. Patients with no or minor limitations are more likely to return to their former sports.

Overall, the results of the outdoor exercise program show general feasibility and a good acceptance among survivors. In 2019, a total of 97 participations took place in 12 activities. However, 5 of the 19 offered activities (26.3%) had to be cancelled because of a lack of participants. Three cancellations can possibly be explained by organizational reasons. The two rowing try-out days, unlike all other activities, did not take place on the weekend but on a Monday afternoon. The participants could therefore have been prevented by school, university, or their working parents. The hiking/climbing weekend camp took place parallel to the scuba diving camp, so participants could only choose one of the two camps. Another reason can be the distance to the location of the event. Due to the low incidence of cancer during childhood, the catchment area for patients is very large and involves long journeys. Ross et al. [29] describes that the majority of childhood cancer survivors prefer a maximum travel time of 10–20 min to participate in an exercise program. The results of the survey will help to further adapt the program to the needs of the participants by, e.g., addressing their barriers to participate. However, it is confining that only participants of the aftercare exercise program were surveyed. Further research is needed, especially to determine the reasons for nonparticipation. For this purpose, patients who have not participated in any offers so far should be surveyed in the future.

The question of how participants became aware of the program shows the importance of the close contact between patients and medical treatment teams. If the employees of the NAOK, the clinical exercise physiologist of the acute hospitals, the physicians, and the mention of the free text response are summarized, it shows that 85.7% became aware of the program directly through treatment personnel. Print media have a share of 42%. Digital media (email and website) were barely mentioned, contributing only 9.5%, with no person mentioning the website. The website did not exist for very long before the survey, which might explain the low recruitment through it. In the meantime, the program was also advertised on several social media channels to reach patients from other clinics as well. The e-mail distribution list represents an important tool for contact with former patients; however, the admission occurs only after the initial contact. Overall, it is evident that personnel and intensive contact with patients is very important in recruiting them to aftercare exercise programs. To attract former patients and patients from other clinics, social media and the dissemination of print media to other clinics are becoming increasingly relevant.

The answers regarding the motivation to participate show that the main argument is the wish to try out a new sport (Figure 3). The participants seem to be very open to outdoor sports in which they have had no previous experience and they would like to see which new sports they could try out after their disease. This fact might be especially relevant for childhood cancer survivors with late effects such as reduced physical capacities or physical limitations such as visual impairments or amputations. Once motivated to participate in a particular sport, it is likely that young people will have the confidence to try other sports as well, until perhaps they find their lifetime sport [30]. Another important argument was to bring a buddy. The support and inclusion of parents, siblings, and peers can positively influence physical activity behavior [31] and also increase numbers of participation and physical activity confidence among pediatric cancer survivors [32,33]. Rosenberg et al. [34] describes that participating in peer-group activities can also improve psychological well-being [35] and coping capabilities [36] for young adults with cancer. It seems evident that the buddy concept lowers the barrier to participate. It also offers the possibility to facilitate sustainability, as the participant and the buddy motivate each other to continue the sport in the long term. Further motivators are physical and psychological factors. On the one hand, participants want to improve their physical well-being and, on the other hand, they want to strengthen their self-confidence. Both parameters can be negatively affected by cancer [5,37,38]. Overall, the responses for motivation show that the children were primarily intrinsically motivated to participate in the offerings. Extrinsic motivators such as the doctor′s recommendation or the cost-free participation were of minor relevance.

Once motivated to participate, the majority of participants are also interested in continuing exercise. The reasons could be the good progress in the respective sports reported by the participants (Figure 5), as well as the special characteristics and settings of outdoor exercising. Progress in the sport and the development of athletic competence are essential for promoting intrinsic motivation and continued participation in sports and physical activity, as seen in healthy school athletes [39]. Due to the special character of outdoor exercising, prior research has revealed positive and restorative effects of support interventions performed in natural environments in women with breast cancer [40]. Furthermore, participants benefit greatly from activities that are physically and mentally challenging without the often-discouraging competition with healthy peers as in most team sports, and thus can provide an opportunity for self-development [30].

Nevertheless, 26.3% of try-out participants and 35.7% of the active weekend camp (somewhat) disagree when asked if they would like to continue the sport beyond the supervised program of NAOK. The reasons may be the nonexistent or unknown sports facilities near home (Figure 5), as, e.g., for water sports activities in a lake, river, or sea the permission to practice sports is required. Furthermore, only 42.1% (try-out day) and 42.9% (active weekend camp) would have the confidence to exercise without the NAOK. Among weekend camp participants, as many as 42.9% answered ‘disagree’ (try-out day: 15.8%) (Figure 5). Free text answers suggest that those answers are mostly attributed to the sport of scuba diving. Participants indicate that there is still too much respect for external instructors, and they believe that other instructors cannot invest as much time, attention, and understanding in the special needs for disabled young cancer survivors. Overall, it is mentioned that trustworthy instructors are very important for the participants. This underlines the need of a team with qualified movement instructors including experts in the respective didactic of sports and pediatric exercise oncology. In addition, it can be seen that for some survivors, a single participation is sufficient to gain self-confidence or to find their sport. However, other participants needed more than one session. Accordingly, future concepts of reintegration in sport clubs or physical education should be individualized and respond to the needs of each participant.

The biggest barriers regarding general sports participation were ‘concerns about not being able to keep up with others’ (Figure 6). This shows the importance of the measures taken by this supervised program to ensure that children, adolescents, and young adults have an opportunity to participate and find access to external exercise options in the long term. Keiser et al. [41] suspects that individual training in the early period after treatment might be even more useful and the level of intensity in training in larger groups cannot be achieved yet.

For this reason, our program initially focuses on trying out different sports individually. After the participants enjoyed and have made progress in the sport, they can gain further experience together in a group with peers at the active weekend camps. The next step will be to gain initial experience in regular sport clubs through offers from cooperation partners. The exercise instructors of the respective partners will have the opportunity to take part in workshops organized by NAOK to learn how to deal with this special cohort. The survivors have the opportunity to deepen their athletic skills, to overcome inhibitions, and to gain self-confidence. The long-term goal is to facilitate the transition from supervised exercise oncology offers to exercise programs in a home setting (sport clubs and physical education).

The limitations of this study are the small sample size and the lack of information on gender or diagnosis in the questionnaire, which was chosen to ensure anonymity in the small group. With a larger group and the additional information, interesting subgroup differences between genders or diagnoses could be investigated in the future.

Overall, the results of this pilot study show that the concept of this outdoor exercise program for children, adolescents, and young adults after cancer treatment was feasible, enjoyable, and safe. Future research is needed to find reasons for the nonparticipation of survivors. For this purpose, participants who have not previously participated in any activities should be surveyed. Findings show that, in particular, the personal contact from clinical staff, cancer patients, and survivors is important, but that new possibilities, such as social media, the website, or the inclusion of other clinics, should also be implemented.

A future goal is to enlarge the recruitment area for the regional NAOK to offer children, adolescents, and young adults from other clinics the opportunity to participate in the exercise program. Further methods need to be developed to facilitate the transition into home settings and to foster lifelong exercise involvement.

## Figures and Tables

**Figure 1 children-09-01117-f001:**
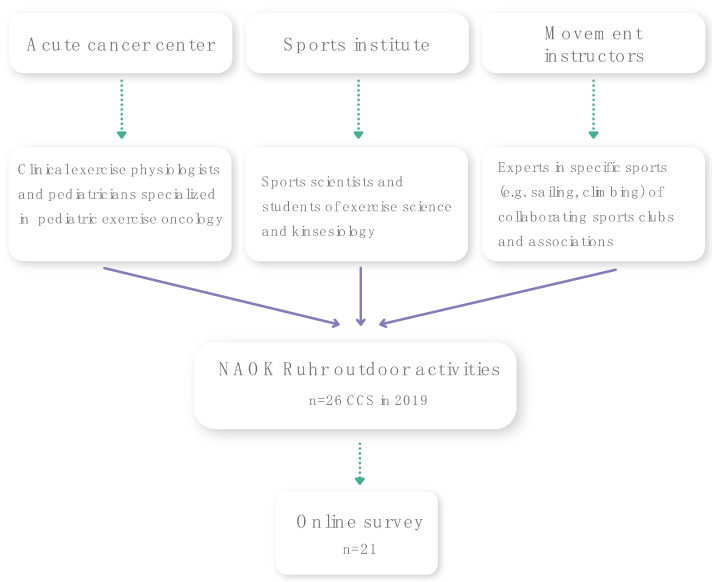
Illustration of the program in the NAOK Ruhr and participation of childhood cancer survivors (CCS) in activities in 2019 and the online survey.

**Figure 2 children-09-01117-f002:**
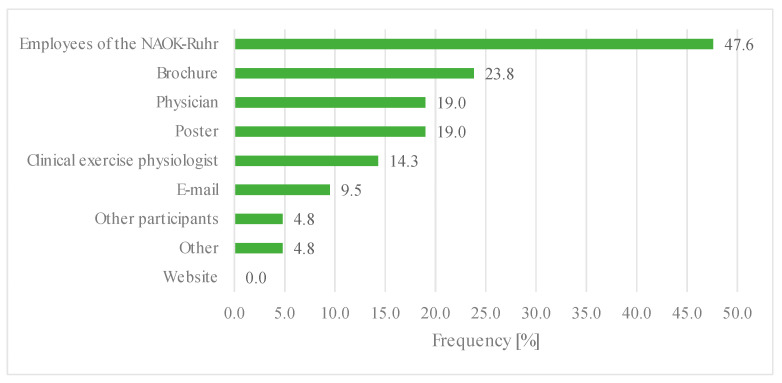
Recruitment ways of participants for the outdoor exercise program.

**Figure 3 children-09-01117-f003:**
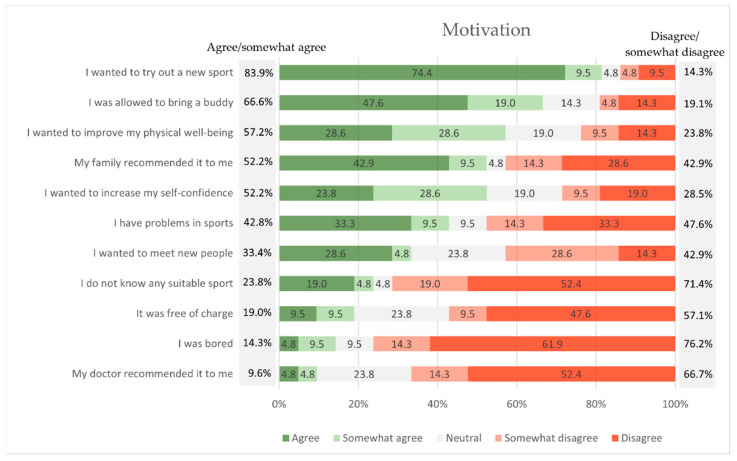
Motivation to exercise participation in the outdoor exercise program.

**Figure 4 children-09-01117-f004:**
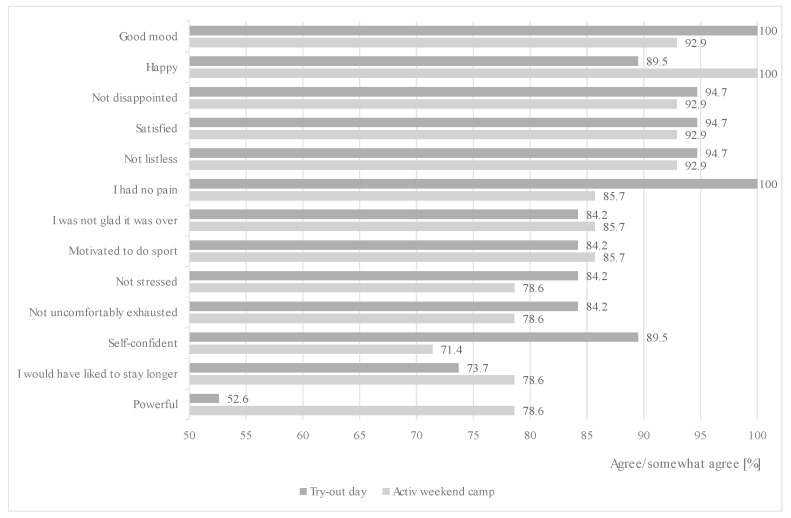
Personal well-being after try-out days and active weekend camps.

**Figure 5 children-09-01117-f005:**
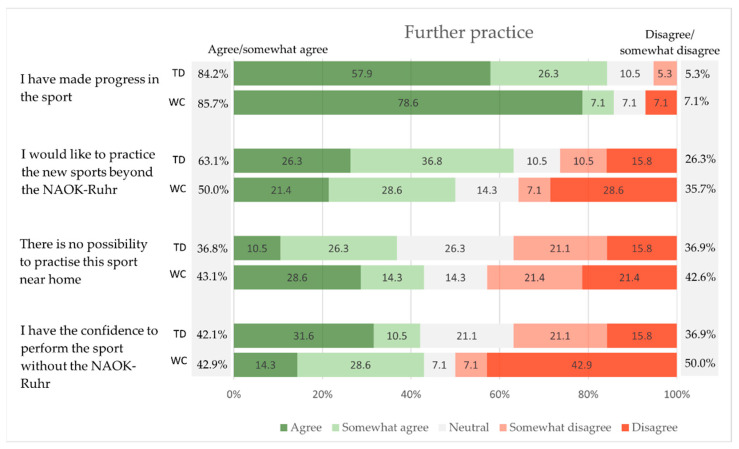
Further practice of the new sports (TD = try-out day, WC = weekend camp).

**Figure 6 children-09-01117-f006:**
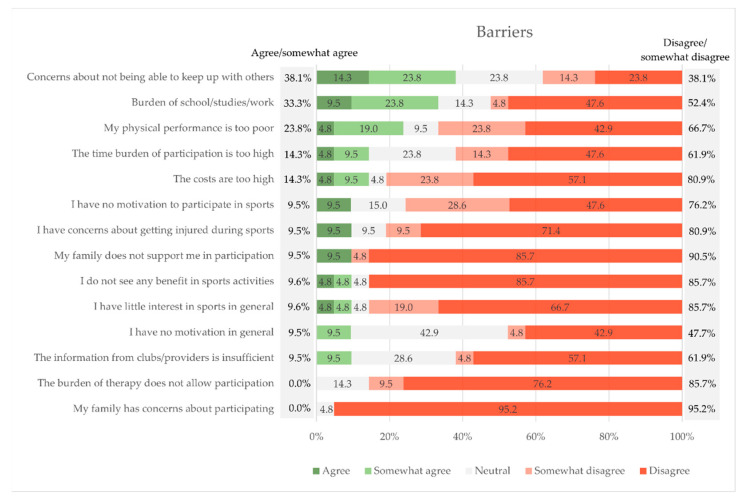
Barriers to general exercise participation.

**Table 1 children-09-01117-t001:** Main demographic and clinical characteristics of the participants.

Participants	N = 26
Gender	m = 14, f = 12
Age (years), mean ± SD (range)	14.6 ± 5.5 (5–29)
Years after diagnosis, mean ± SD (range)	5.3 ± 4.9 (0.7–15.6)
**Types of tumors**	
Leukemia	n = 8 (30.8%)
Brain tumor	n = 6 (23.1%)
Sarcoma	n = 4 (15.4%)
Lymphoma	n = 4 (15.4%)
Germ cell tumor	n = 2 (7.7%)
Nephroblastoma	n = 1 (3.8%)
Neuroblastoma	n = 1 (3.8%)
**Impairments**	
Yes	n = 10 (38.5%)
No	n = 16 (61.5%)
**Kinds of impairments**	
Amputation	n = 2
Megaendoprothesis	n = 2
Vestibular disorder	n = 2
Gait ataxia	n = 1
Polyneuropathy	n = 1
Visual disorder	n = 1
Hearing impairments	n = 1
Cardiac insufficiency	n = 1
Motor disability	n = 1
Seizure tendencies	n = 1

**Table 2 children-09-01117-t002:** Overview of sports and number of participants in 2019.

	Sports (N)		Participants (N)		
Activity	Planned	Completed	Survivors	Buddies	Total
**Try-out days**	16	10	41	35	76
Climbing	2	2	10	9	19
Scuba diving	5	4	14	9	23
Sailing	4	3	10	6	16
Surfing, SUP, kayaking	3	1	7	11	18
Rowing	2	0	0	0	0
**Active weekend camps**	3	2	12	9	21
Scuba diving	1	1	5	4	9
Water sports	1	1	7	5	12
Hiking/Climbing	1	0	0	0	0
**Total activities**	**19**	**12**	**53**	**44**	**97**

## Data Availability

Not applicable.

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
