# Peer review of "Design and Evaluation of an Outdoor Exercise Program for Pediatric Cancer Survivors"

_children, 2022, doi:10.3390/children9081117_

Round 1

Reviewer 1 Report

Thank you for the opportunity to review this interesting paper.

This paper describes a small evaluation of an outdoor exercise program for paediatric cancer survivors. Overall the paper clearly describes the study, provides justification for undertaking the study, clearly presents the results and interprets them appropriately. There are a number of places where the phrasing is unclear. Further details are provided below.

Despite having ‘buddies’ present for the activities, they were not invited to complete surveys. This group could provide a useful control group, especially for understanding barriers and facilitators.

The authors talk in the Introduction about the benefits of outdoor exercise, however participants were not asked about their preferences regarding indoor or outdoor exercise in the survey. This would be interesting to consider.

The Abstract is clearly written overall. Some phrases are a little unclear/awkward such as: ‘exercise concepts’, ‘exercise offers’. Suggest that ‘exercise concepts’ could be ‘exercise programs’ and 'exercise offers’ could be ‘exercise options’.

The Introduction is generally well written.

Lines 49-58. Please ensure it is clear whether the studies described refer to well young people or those with cancer.             

Some terminology or phrasing is unclear or incorrect. For example, the phrase ‘green exercise’ should be clearly defined when first used. It is also unclear if this all outdoor exercise or a subset of outdoor exercise.

Line 143. The description of the questionnaire is unclear. Could the authors please state the total number of questions in each section. The phrase ‘five content areas and several subscales’ is particularly unclear. Please provide clear information on the response scales used in each section and each question or group of questions.

Line 160. The sentence beginning “Before participating…” is unclear. Which data is this referring to? Is this the questionnaires or something else?

In the Discussion, when discussing the past research, it would be helpful if it was clearer which research referred to healthy young people and which to cancer survivors.

The paragraphs in the Discussion are very long. It would aid reading if the sections were broken up more. It would also help if more structure was provided to this section in how topics and sub-topics are presented. Ideas need to be presented in a more cohesive manner.

Line 325. It is not stated in the paper what measures are taken by this program for long term exercise options. It could be helpful if this section was placed in its own paragraph perhaps indicating that it refers to practical or clinical implications/future directions. This section should be separated from the discussion about results.

Lines 314-316. This sentence is unclear. What is meant by ‘still too much respect to be responsive’?

Please comment on the limitation associated with the small sample size. With a larger sample size differences in responses between sub-groups could have been explored.

Figure 1. Please define CCS. How does this relate to the 21 people who did the survey?

Figures 3, 5 and 6. The coloured shading of the bars is difficult to see.

Table 1. The range of participant age goes to 29 years. This is older than adolescence, referring more to young adults. It would be useful if in the introduction, methods and discussion it was mentioned that the program is also for young adults – is there are an upper age limit? The paper is currently inconsistent when describing who the program is for regarding age cohorts (i.e. sometimes referring to adolescents, sometimes young adults etc).

Table 2. Does Diving refer to diving from a diving board or underwater diving (e.g. scuba diving)?

There are small grammatical errors throughout, such as:

Line 38. Again, not clear what is meant by ‘exercise concepts’.

Line 41. ‘are’ not ‘is’

Line 42 – what is the relevance of ‘political changes’?

Line 43 – ‘number’ not ‘amount’

Line 61 – ‘Participation by’ not ‘Participations of’

Reviewer 2 Report

All in all this is a fine little project and as such suitable for publication. I suggest you go through the results one more time and check if there were differences in opinion between boys and girls or different age groups. Also it could be of importance to note the time of the original diagnosis. 

The study population is very small and as such I don't expect much quantitative analyses to be done for any subgroups - maybe just a note "boys tended to appreciate.." or "older children as well as children who had recovered from their cancer for more than xx years tended to be..." 
